# Mesoscale Temporal Wind Variability Biases Global Air–Sea Gas Transfer Velocity of CO2 and Other Slightly Soluble Gases

**Yuanyuan Gu [1,2,*], Gabriel G. Katul [3,4] and Nicolas Cassar [1,5]**

[1] Division of Earth and Ocean Sciences, Nicholas School of the Environment, Duke University, Durham, NC 27708, USA; Nicolas.Cassar@duke.edu

[2] College of Oceanography, Hohai University, Nanjing 210098, China

[3] Nicholas School of the Environment, Box 90328, Duke University, Durham, NC 27708, USA; gaby@duke.edu

[4] Department of Civil and Environmental Engineering, Duke University, Durham, NC 27708, USA

[5] CNRS, Univ Brest, IRD, Ifremer, LEMAR, F-29280 Plouzané, France

[*] Correspondence: yuanyuan.gu@duke.edu

**Abstract:** The significance of the water-side gas transfer velocity for air–sea CO2 gas exchange (k) and its non-linear dependence on wind speed (U) is well accepted. What remains a subject of inquiry are biases associated with the form of the non-linear relation linking k to U (hereafter labeled as f(U), where f(.) stands for an arbitrary function of U), the distributional properties of U (treated as a random variable) along with other external factors influencing k, and the time-averaging period used to determine k from U. To address the latter issue, a Taylor series expansion is applied to separate $f(U)$ into a term derived from time-averaging wind speed (labeled as $\langle U \rangle$, where $\langle . \rangle$ indicates averaging over a monthly time scale) as currently employed in climate models and additive bias corrections that vary with the statistics of U. The method was explored for nine widely used f(U) parameterizations based on remotely-sensed 6-hourly global wind products at 10 m above the sea-surface. The bias in k of monthly estimates compared to the reference 6-hourly product was shown to be mainly associated with wind variability captured by the standard deviation $\sigma_U$ around $\langle U \rangle$ or, more preferably, a dimensionless coefficient of variation $I_u = \sigma_U / \langle U \rangle$. The proposed correction outperforms previous methodologies that adjusted k when using $\langle U \rangle$ only. An unexpected outcome was that upon setting $I_u^2 = 0.15$ to correct biases when using monthly wind speed averages, the new model produced superior results at the global and regional scale compared to prior correction methodologies. Finally, an equation relating $I_u^2$ to the time-averaging interval (spanning from 6 hours to a month) is presented to enable other sub-monthly averaging periods to be used. While the focus here is on CO2, the theoretical tactic employed can be applied to other slightly soluble gases. As monthly and climatological wind data are often used in climate models for gas transfer estimates, the proposed approach provides a robust scheme that can be readily implemented in current climate models.

**Keywords:** carbon dioxide; gas transfer velocity; time-averaging; wind speeds

## 1. Introduction

Describing air–sea flux of long-lived greenhouse gases such as carbon dioxide (CO2) is of significance for assessing the global carbon cycle and its relation to climate. In climate models, the water-side air–sea flux (F, mol m$^{-2}$ y$^{-1}$) is commonly determined using a bulk expression

$$F = K_0 \, k \, \Delta pCO_2 \qquad (1)$$

where k is the gas transfer velocity (cm h$^{-1}$), $K_0$ is the gas solubility (mol L$^{-1}$ atm$^{-1}$) in water that is a function of sea surface temperature (SST) and salinity, and $\Delta pCO_2$ is the difference in partial pressure of pCO2 between water and air (atm). The dominant factors

determining k are governed by a number of physical processes primarily, but not exclusively, associated with wind speed U . For this reason, k is operationally parameterized as a non-linear function of U set at a reference height of 10 m. For comparison purposes, the general formulations (common ones listed in Table 1) take the form of

$$k = (Sc/660)^{-1/2} f(U) \qquad (2)$$

where $f(U)$ (cm h$^{-1}$) is a non-linear function of wind speed U, also known as the gas transfer velocity $k_{660}$ normalized to the dimensionless molecular Schmidt number (Sc) for $CO_2$ in seawater at 20 °C (Sc = 660). The function $f(U)$ may be quadratic, cubic, or even a higher-order polynomial, and Sc ($\gg$1) is the ratio of the kinematic viscosity (m$^2$ s$^{-1}$) and the molecular diffusion coefficient (m$^2$ s$^{-1}$) of $CO_2$ or other gases in seawater. For this reason, Equation (2) is routinely used for slightly soluble gases. The $f(U)$ can also be derived using turbulent transport theories [1–5], bubbles [6,7], and wave-breaking mechanics [8–10]. However, $f(U)$ cannot be viewed as linking k to an instantaneous U at a point; rather, $f(U)$ must emerge as an approximation to macroscopic equations derived by averaging gas transfer over space and time scales (analogous to a closure model for turbulent fluxes in Reynolds-averaged Navier–Stokes equations). The spatial scales must be much larger than the largest eddy or wave length impacting gas exchange, whereas the time scales must be sufficiently long to accommodate the effects of turbulent fluctuations (i.e., ensemble of many eddy-turnover times) or wave formation and subsequent breaking, but short enough to resolve mesoscale variations in U . This interval is commensurate with hourly time scales and coincides with time scales associated with the well-known spectral gap in the atmosphere [11]. Fourier power spectra of wind time series sampled from fractions of seconds (turbulent scales) to years support the occurrence of a "gap" in the squared Fourier amplitudes separating mesoscales (longer than few hours) from micro-scales (smaller than minutes). This gap forms the basis of separating U into a micro-scale contribution whose effects on k are to be averaged out and captured by $f(U)$ and a meso-scale or longer (i.e., larger than hours) contribution [12].

Based on gas transfer velocity parameterizations, modelling and observation-based estimates of the global oceanic $CO_2$ sink vary significantly from –1.18 to –3.1 Pg C yr$^{-1}$ (negative referring to net flux of $CO_2$ into the ocean) [13–18]. The range in these estimates reflects different time periods and uncertainties. Uncertainties result from using various data products, methodological uncertainties in k parameterizations, the relative sparsity of $CO_2$ data coverage in time and space, and thermal and haline effects [19–25].

Additionally, temporal averaging of wind data substantially contributes to uncertainties in global F estimates due to the non-linearity in $f(U)$ [26–28] and frames the scope of the work here. As an example, if wind blows half the time at a speed of 4 m s$^{-1}$ and the other half of the time at 16 m s$^{-1}$ (solid points on the curves), k values estimated from the mean wind speed of 10 m s$^{-1}$ are biased low by 11.2 and 30.6 cm h$^{-1}$ relative to the true k (circles on dash lines) for the quadratic and the cubic relations, respectively. Quadratic and cubic equations are taken from [29] and [30], respectively (Figure 1). Long-term averaged (monthly or longer) wind speeds underestimate gas exchange by 25% and by 50% for quadratic and cubic $f(U)$, respectively [31]. Such known biases can be handled by: (i) using wind speeds with short temporal intervals (i.e., 6 hours) or (ii) applying correction factors when averaging over longer intervals (e.g., monthly) to mitigate these expected biases [26–28]. Because using short-term wind speeds (e.g., 6-hourly wind products) globally to evaluate $f(U)$ is computationally expensive now and in the foreseeable future, bias-corrected methodologies are gaining attention. However, reported biases in k are still pronounced even after applying current correction factors, thereby motivating the development of other approaches. The time is ripe to begin exploring such bias corrections to existing gas exchange formulations given the availability of satellite-based wind products at 6-hourly temporal resolution.

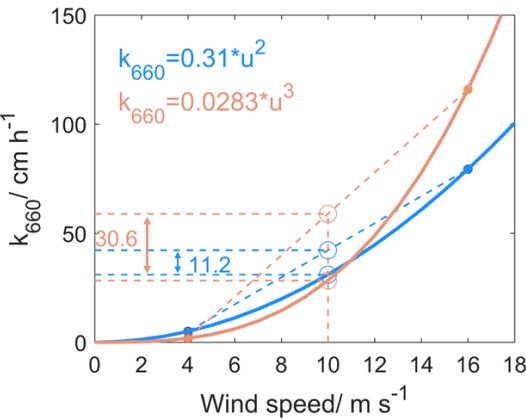

**Figure 1.** Conceptual diagram representing the bias in gas transfer velocity (k) estimates associated with averaging wind speed variability (adapted from [32]). The quadratic and cubic relations are in blue and orange, respectively.

In this study, we compare various published corrections and propose a new method that we test globally and regionally for any function f(U). The findings here apply for any slightly soluble gas for which its f(U) is known. For a more accurate correction, wind variance should also be supplied to correct monthly k. In the absence of such information, the work here suggests a constant squared coefficient of variation can be used ($I_{u^2} = 0.15$). The manuscript is organized as follows: the datasets and data processing, review of current correction methods, and the proposed new correction method are presented in Section 2. In Section 3, this new method is applied to 29 years of data to obtain the corrected gas transfer velocity and is evaluated by comparing the newly corrected k to results from earlier models and studies. A summary and concluding remarks are presented in Section 4.

**Table 1.** The f(U) parameterizations used in estimating gas transfer velocity for $CO_2$ (same expressions can be used for other slightly soluble gases [33]). The f(U) formulations developed from long-term wind speeds (i.e., monthly) were not considered here. In Serial No.9, because the three equations are identical in form with a small difference in their $\alpha$ coefficients, we use the expression f(U) = 0.251 $U^2$ [25] in the following analysis as a representative equation of all three models.

| Serial No. | Reference | f(U) Parameterization for $CO_2$ $f(U) = k \, (Sc/660)^{1/2}$ |
|:---:|:---:|:---:|
| 1 | Wanninkhof (1992) [29] | $0.31U^2$ |
| 2 | Wanninkhof and McGillis (1999) [30] | $0.0283U^3$ |
| 3 | Nightingale et al. (2000) [34] | $0.222U^2 + 0.333U$ |
| 4 | McGillis et al. (2001) [35] | $0.026U^3 + 3.3$ |
| 5 | McGillis et al. (2004) [36] | $0.014U^3 + 8.2$ |
| 6 | Weiss et al. (2007) [37] | $0.365U^2 + 0.46U$ |
| 7 | Wanninkhof et al. (2009) [38] | $0.011U^3 + 0.064U^2 + 0.1U + 3$ |
| 8 | Prytherch et al. (2010) [39] | $0.034U^3 + 5.3$ |
| 9 | Ho et al (2006) [40], Sweeney et al. (2007) [41], Wanninkhof (2014) [33] | $\alpha U^2$ (where $\alpha = 0.266/0.27/0.251$) |

## 2. Data and Methods

### 2.1. Data and Data Processing

The U and sea surface temperature (SST) data from 1990 to 2018 are used to compute globally averaged k. The 6-hourly and averaged monthly cross-calibrated multiplatform

CCMP V2.0 wind data at the 0.25° × 0.25° grid were obtained from the Remote Sensing Systems described elsewhere [42]. The gridded CCMP V2.0 wind products are produced from a combination of satellite (Version-7 remote sensing system radiometer wind speeds, QuikSCAT, and ASCAT scatterometer wind vector wind), moored buoy, and model wind data. The high-resolution CCMP captures wind variability well and is relatively bias free compared to in situ estimates [43,44].Therefore, the CCMP data is often used to parameterize the gas transfer velocity coefficient and in estimating k [25,33,43,45]. The 6-hourly and monthly SST data at the spatial resolution of 0.25° × 0.25° were derived from the European Centre for Medium-Range Weather Forecasts (ECMWF) fifth generation ERA5 reanalysis product described elsewhere [46]. The wind speeds and SST are linearly interpolated onto a spatial resolution of 0.5° × 0.5°. Within this grid, the statistics of $U$ are assumed to be planar homogeneous. Additionally, both $U$ and SST data were linearly interpolated to 5° × 5° to evaluate the effect of spatial resolution on k estimates. Nine commonly used $f(U)$ parameterizations (Table 1) were applied to estimate globally averaged gas transfer velocity for $CO_2$. The Sc for $CO_2$ is a function of SST and is determined using a standard formulation [33].

## 2.2. Review of Prior Correction Methods for the Time-Average Bias

Correction factors are routinely applied to account for biases in k estimates associated with time-averaging of wind speeds. One commonly used method is based on a Reynolds decomposition into a well-defined mean $\langle U \rangle$ (e.g., monthly) and fluctuations $U'$ (on the scales of hours) around this average so that $U = \langle U \rangle + U'$ with $\langle U' \rangle = 0$. When $f(U) = aU^2$, $f(\langle U \rangle) = a\langle U \rangle^2$ and $\langle f(U) \rangle = \langle aU^2 \rangle$. Therefore, the $\langle f(U) \rangle$ can be evaluated as

$$\langle f(U) \rangle = a\langle(\langle U \rangle + U')^2\rangle = a\big(\langle U \rangle^2 + 2\langle U \rangle\langle U' \rangle + \langle U'^2 \rangle\big) = a\langle U \rangle^2\left[1 + \left(\frac{\sigma_U}{\langle U \rangle}\right)^2\right] \tag{3}$$

$$= f(\langle U \rangle)[1 + (\sigma_U/\langle U \rangle)^2]$$

where $\sigma_U$ is the standard deviation, $I_u = (\sigma_U/\langle U \rangle)$ is as before the coefficient of variation, and the sought correction (as a ratio or bias) can be expressed as

$$\frac{\langle f(U) \rangle}{f(\langle U \rangle)} = (1 + I_u{}^2) \text{ and } \langle f(U) \rangle - f(\langle U \rangle) = a\sigma_U^2 \tag{4}$$

Clearly, this correction depends on the non-linearity of $f(U)$ [32,47]. With available 6-hourly wind speed ($U_{6\,hour}$) data, other widely used multiplier corrections [26–28,31,40,45,48–52] for the quadratic formulation is expressed in this form as

$$R_2 = \langle U_{6\,hour}^2 \rangle / \langle U_{6\,hour} \rangle^2 \tag{5}$$

and for the cubic formulation

$$R_3 = \langle U_{6\,hour}^3 \rangle / \langle U_{6\,hour} \rangle^3 \tag{6}$$

These $R_2$ and $R_3$ corrections can be obtained empirically or derived analytically when assuming the distributional properties of $U$ for meso-scale (and longer) variations [26–28,31,40,45,48–52]. A Rayleigh distribution, which is commonly used in the evaluation of $R_2$ and $R_3$ [26] arises when the magnitude of the wind velocity is analyzed in two dimensions (usually in the plane parallel to the water surface) (see Text S1 in Supplementary). Assuming that each component is uncorrelated and normally distributed with equal variance in each of the two directions (i.e., planar homogeneous air flow), the overall wind vector magnitude is characterized by a Rayleigh distribution (i.e., a special form of Chi-squared). These $R_2$ and $R_3$ corrections are also simplified using zonally averaged profiles [26,27]. Globally and regionally, the $R_2$ ranges from 1.12 to 1.26 whereas the $R_3$ ranges from 1.35 to 2.17 [26,28,48,53].

*2.3. Proposed Correction Based on Taylor Series Expansions*

For a wind-only related formulation, $f(U) \propto U^n$ ($n > 1$) and upon space-time averaging yields $\langle U^n \rangle \neq \langle U \rangle^n$. To assess biases arising from setting $\langle f(U) \rangle = f(\langle U \rangle)$, a Taylor series expansion of any $f(U)$ form around the space-time averaged value $f(\langle U \rangle)$ are now introduced and given by

$$f(U) = f(\langle U \rangle) + \frac{df}{dU}\bigg|_{\langle U \rangle}(U - \langle U \rangle) + \frac{1}{2!}\frac{d^2f}{dU^2}\bigg|_{\langle U \rangle}(U - \langle U \rangle)^2 + \frac{1}{3!}\frac{d^3f}{dU^3}\bigg|_{\langle U \rangle}(U - \langle U \rangle)^3 + \cdots \tag{7}$$

Applying the space-time averaging operation $\langle . \rangle$ term by term in Equation (7) yields

$$\langle f(U) \rangle = f(\langle U \rangle) + \frac{df}{dU}\bigg|_{\langle U \rangle}\langle U - \langle U \rangle \rangle + \frac{1}{2}\frac{d^2f}{dU^2}\bigg|_{\langle U \rangle}\langle (U - \langle U \rangle)^2 \rangle$$
$$+ \frac{1}{6}\frac{d^3f}{dU^3}\bigg|_{\langle U \rangle}\langle (U - \langle U \rangle)^3 \rangle + \cdots \tag{8}$$

This expression can be arranged as

$$\langle f(U) \rangle = f(\langle U \rangle) + k_b,$$

$$\text{with } k_b = \underbrace{\langle f(U) \rangle - f(\langle U \rangle)}_{\text{Term 1}} = \underbrace{\frac{1}{2}\frac{d^2f}{dU^2}\bigg|_{\langle U \rangle}\sigma_U^2 + \frac{1}{6}\frac{d^3f}{dU^3}\bigg|_{\langle U \rangle}\langle (U - \langle U \rangle)^3 \rangle \cdots}_{\text{Term 2}}, \tag{9}$$

where $k_b$ is the sought bias and $\sigma_U^2$ is the variance in wind speed around $\langle U \rangle$. In this expression, all derivatives of $f(U)$ are presumed to be known (e.g., Table 1 or other physics-based formulation) and are being evaluated at $\langle U \rangle$. When $n = 1$, all the derivative terms are identically zero and $k_b = 0$. General expression for a quadratic relation ($n = 2$) $f(U)$ can be expressed as

$$f(U) = aU^2, \tag{10}$$

applying Equation (9) to $f(U)$ results in

$$\langle f(U) \rangle = a\langle U \rangle^2 + k_b, \text{ with } k_b = a\,\sigma_U^2, \tag{11}$$

suggesting an additive correction (i.e., bias) that only varies with $\sigma_U^2$. This expression is consistent with [47], though the approach taken here is more general. For the cubic relations ($n = 3$),

$$f(U) = a\,U^3 + bU^2 + dU + e \tag{12}$$

and this results in

$$\langle f(U) \rangle = a\,\langle U \rangle^3 + b\langle U \rangle^2 + d\langle U \rangle + e + k_b,$$
$$\text{with } k_b = 3a\langle U \rangle\sigma_U^2 + b\sigma_U^2 + a\langle (U - \langle U \rangle)^3 \rangle. \tag{13}$$

In this case, the skewness of $U$ as estimated with the Fisher–Pearson coefficient of skewness ($Sk = \langle (U - \langle U \rangle)^3 \rangle / \sigma_U^3$) is also required. In summary, the newly proposed corrections and the three prior correction methods listed in Table 2 are used to adjust for the bias.

**Table 2.** Summary of all the correction methodologies for $CO_2$ and other gases.

| Method | Reference | Correction | Correction Details |
|--------|-----------|------------|--------------------|
| 1 | This study | $k_b$ from Equation (11) (for quadratic relations) and Equation (13) (for cubic relations) are added to $f(\langle U \rangle)$ to estimate the corrected k. | Grid-by-grid spatially multi-year mean $k_b$ |
| 2 | This study | | A simplified method using overall averaged value of $k_b$ to fix the bias. |
| 3 | Wanninkhof (2002) [26] | (1) The corrected k with multiplier correction $R_2$ (Equation(5)) for the quadratic parameterization is in the form of $f(\langle U \rangle) = a\, R_2 \langle U \rangle^2$, (2) For the cubic relation with multiplier correction $R_3$ (Equation(6)), the corrected $f(U)$ is expressed as $f(\langle U \rangle) = a\, R_3 \langle U \rangle^3 + b\langle U \rangle^2 + d\langle U \rangle + e$ | Assuming a Rayleigh distribution of the 6-hourly wind speeds, $R_2 = \frac{\Gamma(2)}{[\Gamma(3/2)]^2} = 1.27$ and $R_3 = \frac{\Gamma(5/2)}{[\Gamma(3/2)]^3} = 1.91$ (See Text S1 in Supplementary for details). |
| 4 | Jiang et al. (2008) [28] | | Global averaged multiplier correction factors $R_2$ and $R_3$ are estimated using the measured 6-hourly wind speed with $R_2 = 1.23$ and $R_3 = 1.78$. |
| 5 | Fangohr et al. (2008) [27] | | Zonal averaged $R_2$ and $R_3$ are used. Large gradients in zonal $R_2$ and $R_3$ are because of the large zonal gradients in wind variance (Figure S1). |

## 3. Results

### 3.1. Bias in k Induced by Averaging of Wind Data

Globally averaged k for $CO_2$ at various temporal and spatial resolutions were assessed for the parameterizations of $f(U)$, listed in Table 1. The k computed from maximum spatial ($0.5° × 0.5°$) and temporal resolution (6-hourly) products were used as a reference to illustrate the deviation of k in percentage (Figure 2 and Table S1). As expected, the monthly k underestimates k for all parameterizations. The absolute biases induced by time-averaging wind speed (~10–28% range) are more significant for cubic relations (Serial NO. (2), (4), (5), (7), and (8)) than for quadratic relations (Serial NO. (1), (3), (6), and (9)), with a comparable magnitude in biases at both spatial resolutions of $0.5° × 0.5°$ and $5° × 5°$. In contrast, uncertainties due to differential spatial resolutions are negligible (less than 1%) at the temporal resolutions of both 6 hours and a month (Figure 2). For this reason, we only focus hereafter on the uncertainty induced by differential temporal resolution of wind speed data, though the method can be applied to any type of averaging [28]. As expected, k substantially varies with the choice of $f(U)$ being used (Figure S2). Undoubtedly, the mechanisms constraining gas transfer velocity must be explored, but this issue is beyond the scope of the present work.

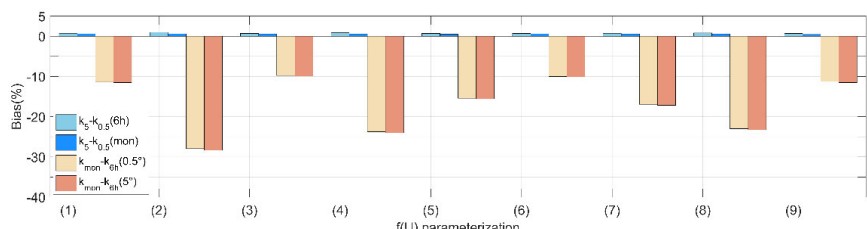

**Figure 2.** Bias in k of $CO_2$ due to wind speeds at varying spatial resolutions ($0.5° × 0.5°$ and $5° × 5°$) for 6-hourly and monthly gas transfer velocity (k), and temporal bias in k (6 hourly and monthly) at the spatial resolution of $0.5° × 0.5°$ and $5° × 5°$. The $k_{mon}$ and $k_{6h}$ are gas transfer velocities averaged over all k values estimated from monthly and 6-hourly wind speed records, respectively. $k_{5°}$ and $k_{0.5°}$ are gas transfer velocities averaged over all k values estimated from $5°$ and $0.5°$ wind speed, respectively. The bias is estimated as $\triangle k*100/k_{6h,0.5°}$ ($k_{6h,0.5°}$ is k at the resolution of 6-hourly and $0.5° × 0.5°$).

### 3.2. Assessment of the "Bias Correction Model"

This new "bias correction model" was applied to all the nine parameterizations of k, as shown in Table 1. Measured bias in f(U) (term 1 in Equation(9), Figure S3) and bias $k_b$ from the new model (term 2 in Equation(9), Figure S4) were estimated. In term 1, the 6-hourly space-time product was used to evaluate $\langle f(U) \rangle$ and the monthly space-time product was used to evaluate $f(\langle U \rangle)$. Overall, the proposed model reproduces the bias between 6-hourly k and monthly k (Figure 3). Spatially, the differences in the first term and the second term are negligible in quadratic parameterizations ((1), (3), (6), and (9) in Figure S5). In contrast, for the cubic relations such as the k parameterizations of (2), (4), (5), (7), and (8), the differences are small, lower than 0.6 cm h$^{-1}$ (Figure 3). The higher values in the mid and high latitude of the northern hemisphere (Figure S5) might be associated with large variability in wind speed within a month (Figure S1) due to the occurrence of synoptic high wind events in these regions [54].

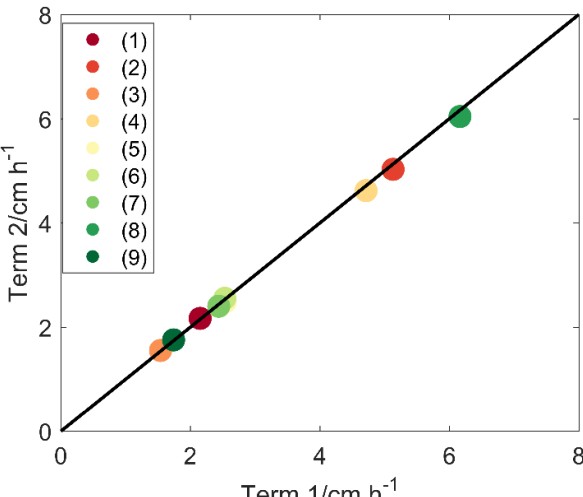

**Figure 3.** Mean bias in gas transfer velocity (k) for $CO_2$ estimated from term 1 (measured bias in f(U)) and term 2 (bias correction $k_b$ from new model) of Equation (9) over the period spanning 1990 to 2018 for the parameterizations presented in Table 1.

### 3.3. Comparison of Correction Methods

The proposed new bias correction was compared to common correction methodologies. We tested the new corrections presented in Equation (11) and Equation (13) based on two methods. In method 1, the correction was estimated using the annual mean grid-by-grid $k_b$ (i.e., with grid-by-grid $\sigma_U^2$, $\langle U \rangle$ and $\langle (U - \langle U \rangle)^3 \rangle$). In method 2, the correction was the averaged $k_b$ (i.e., with averaged $\sigma_u$-related terms) (Table 2). As $\sigma_u$ and $\langle U \rangle$ both increase in time, the squared coefficient of variation $I_u^2 = (\sigma_U / \langle U \rangle)^2$ is a more "conserved" parameter in time with a slowly decreasing (insignificant) trend of 0.002 dec$^{-1}$ and an average of 0.15 (Figure 4). Thus, for a constant $I_u^2 = 0.15$ in method 2, the $k_b$ of quadratic relations (Equation (11)) can be arranged as

$$k_b = a \langle U \rangle^2 \left( \frac{\sigma_u}{\langle U \rangle} \right)^2 = a \langle U \rangle^2 I_u^2 = 0.15 a \langle U \rangle^2 \tag{14}$$

To simplify further, only the first term in Equation (12) is taken for cubic expressions, so that

$$\langle f(U) \rangle = a \langle U \rangle^3 + k_b \tag{15}$$

Evidently, $k_b$ is also a function of $I_u^2$ for cubic expressions given as

$$k_b = 3a \langle U \rangle^3 \left( \frac{\sigma_u}{\langle U \rangle} \right)^2 = 3a \langle U \rangle^3 I_u^2 = 0.45a \langle U \rangle^3 \quad (16)$$

With constant $I_u^2 = 0.15$, Equation (14) and Equation (16) can be used to approximate unbiased f(U), and k considering differences in Sc for $CO_2$ and other gases (see [33] for Sc of other gases). The same f(U) parameterization for all slightly soluble gases may not be realistic for gases with differing solubilities [55–57], but this inquiry is better kept for the future.

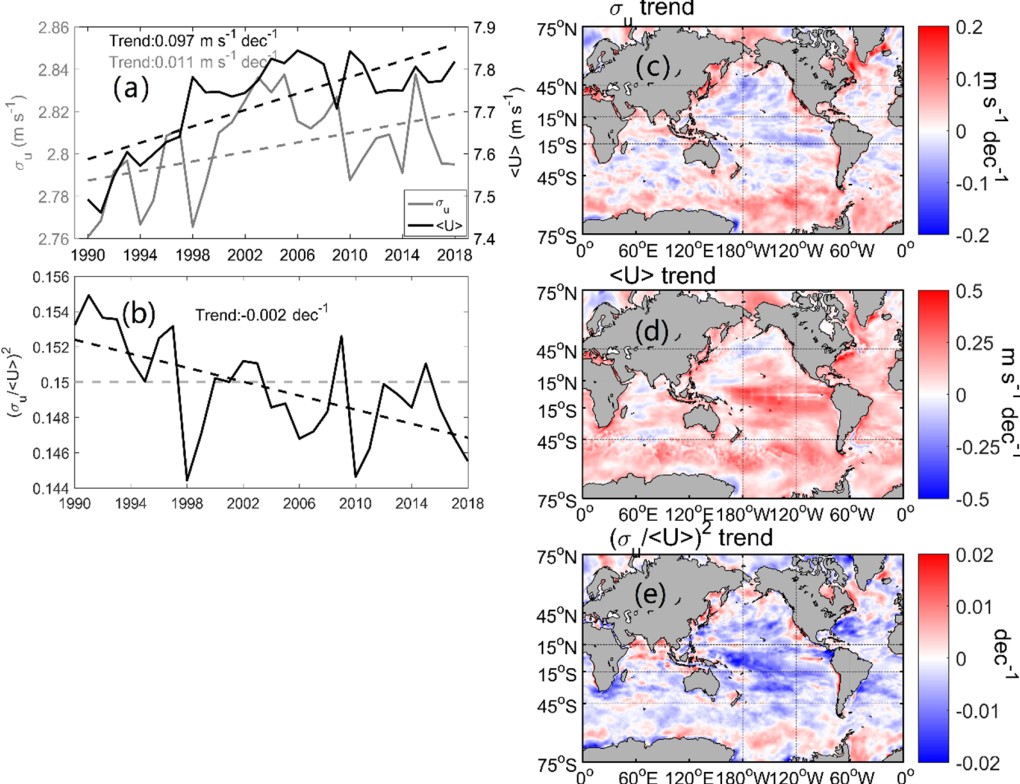

**Figure 4.** Left panel: time series of global (**a**) monthly averaged wind speed $\langle U \rangle$ (in black) and standard deviation ($\sigma_u$, in grey) around $\langle U \rangle$, (**b**) monthly squared coefficient of variation $I_u^2 = (\sigma_u / \langle U \rangle)^2$ from 1990 to 2018 (note the small variations along the ordinate axis). The black and the grey dashed lines in (**b**) indicate the long-term trend (0.002 dec$^{-1}$) and average ($I_u^2 = 0.15$), respectively. Right panel: spatial distribution of (**c**) trends in the wind speed standard deviation ($\sigma_u$) around $\langle U \rangle$, (**d**) monthly averaged wind speed $\langle U \rangle$, and (**e**) monthly squared coefficient of variation $I_u^2 = (\sigma_u / \langle U \rangle)^2$ from 1990 to 2018.

The newly proposed corrections were compared to three commonly used methods (listed in Table 2) when adjusting wind variability-induced bias in k for wind speeds sampled at a resolution of 0.5° × 0.5° spatially and temporally monthly. Taking $CO_2$ as an example, globally averaged corrected k values were calculated for all the nine parameterizations, and biases in corrected k are estimated in reference to the 6-hourly k.

As expected, deviations of the corrected k from the reference 6-hourly k are significantly reduced using the new approaches (Figure 5). The largest absolute biases yielded by methods 1 and 2 are only 0.6% and 4.46%, respectively. In contrast, the range of absolute biases generated by the other three methods vary from 3.5% to 28% (Table S2). It is worth noting that the bias induced by the time averaging of SST values is negligible (small difference in 6-hourly and corrected k after applying correction only associated with wind variance) due to the small variance of SST (Figure 6) and the smaller effect of SST on k for all parameterizations (Table 3). From the perspective of zonal distribution,

the magnitude of the corrected k using method 1 agrees well with the 6-hourly k (Figure 7a). Though the simpler method 2 appears to overestimate k at mid and high latitude regions (within 0–30°N and 30°–60°N), the overall corrected k is generally consistent with the 6-hourly k (especially in the Southern hemisphere), with a smaller root-mean-square errors (RMSE) between corrected k and the 6-hourly k compared to methods 3, 4, and 5 (Figure 7b).

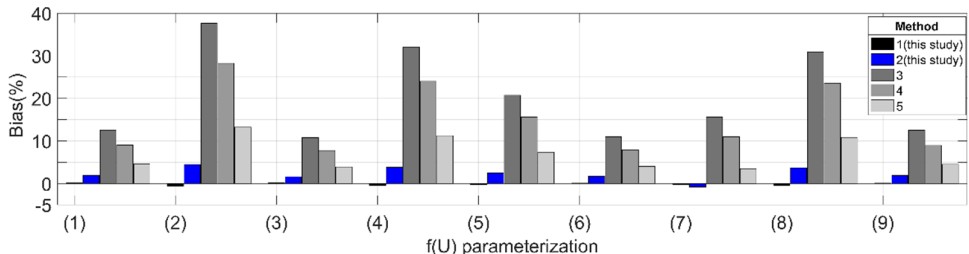

**Figure 5.** Difference in 6-hourly k and corrected k for $CO_2$ applying five correction methodologies in reference to the 6-hourly k (in %) for k parameterizations listed in Table 1. The bias is estimated as $\triangle k*100/k_{6h,0.5°}$.

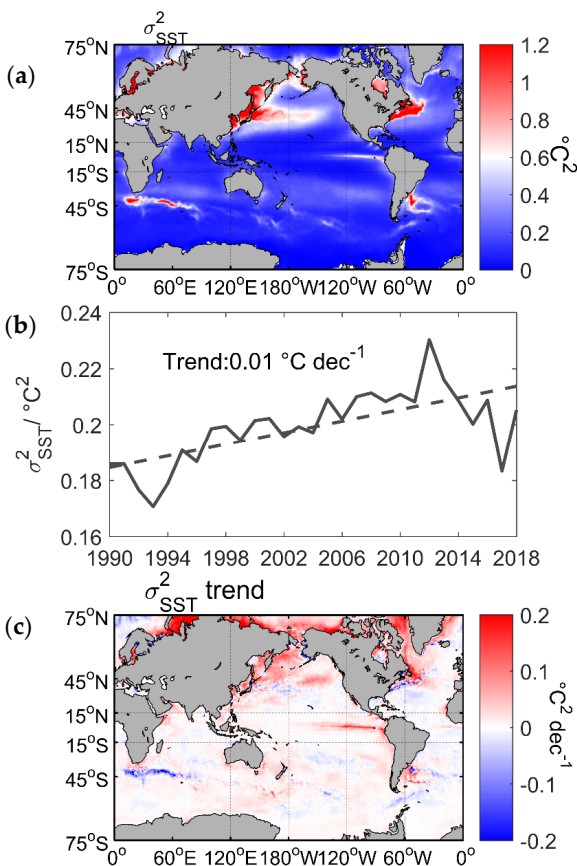

**Figure 6.** (**a**) Spatial distribution of averaged variance of sea surface temperature (SST) ($\sigma_{SST}^2$) around monthly averaged ⟨SST⟩; (**b**) Time series of annual averaged variance of SST ($\sigma_{SST}^2$) around monthly averaged ⟨SST⟩, the dashed line indicates the long-term trend; (**c**) Spatial pattern of trend in averaged variance of SST ($\sigma_{SST}^2$) around monthly averaged ⟨SST⟩ from 1990 to 2018.

**Table 3.** Parameters used to run scenarios of imposed changes in wind speed (U) and sea surface temperature (SST) and their effects on the gas transfer velocity (k) for the nine k parameterizations featured in Table 1. The starting values of U and SST were set to 6.48 m s⁻¹ and 13.73 °C, respectively, according to their climatological global mean. The sensitivities of k to U and SST were assessed from the ratio of the percentage change in k (Y) to percentage change in each factor (X) using the equation: sensitivity = $(\Delta Y/Y)/(\Delta X/X)$.

| Serial NO | Starting Value | | Imposed Change | | | | | | Imposed Change | | | | | |
|---|---|---|---|---|---|---|---|---|---|---|---|---|---|---|
| | | | U | | | SST | | | U | | | SST | | |
| | $U$ (m s⁻¹) | SST (°C) | 2% | 4% | 8% | 2% | 3% | 4% | 2% | 4% | 8% | 2% | 3% | 4% |
| | | | $\Delta k$ | | | | | | k Sensitivity | | | | | |
| 1 | | | 0.49 | 1.00 | 2.04 | 0.09 | 0.14 | 0.18 | 2.02 | 2.04 | 2.08 | 0.38 | 0.38 | 0.38 |
| 2 | | | 0.47 | 0.95 | 1.98 | 0.06 | 0.09 | 0.11 | 3.06 | 3.12 | 3.25 | 0.38 | 0.38 | 0.38 |
| 3 | | | 0.39 | 0.79 | 1.61 | 0.08 | 0.12 | 0.16 | 1.84 | 1.85 | 1.89 | 0.38 | 0.38 | 0.38 |
| 4 | | | 0.43 | 0.88 | 1.82 | 0.07 | 0.11 | 0.15 | 2.19 | 2.24 | 2.32 | 0.38 | 0.38 | 0.38 |
| 5 | 6.84 | 13.73 | 0.23 | 0.47 | 0.98 | 0.08 | 0.12 | 0.16 | 1.08 | 1.10 | 1.15 | 0.38 | 0.38 | 0.38 |
| 6 | | | 0.42 | 0.86 | 1.75 | 0.08 | 0.12 | 0.16 | 2.02 | 2.04 | 2.08 | 0.38 | 0.38 | 0.38 |
| 7 | | | 0.43 | 0.87 | 1.77 | 0.08 | 0.12 | 0.16 | 2.02 | 2.04 | 2.08 | 0.38 | 0.38 | 0.38 |
| 8 | | | 0.30 | 0.60 | 1.24 | 0.06 | 0.10 | 0.13 | 1.72 | 1.74 | 1.80 | 0.38 | 0.38 | 0.38 |
| 9 | | | 0.40 | 0.81 | 1.65 | 0.07 | 0.11 | 0.15 | 2.02 | 2.04 | 2.08 | 0.38 | 0.38 | 0.38 |

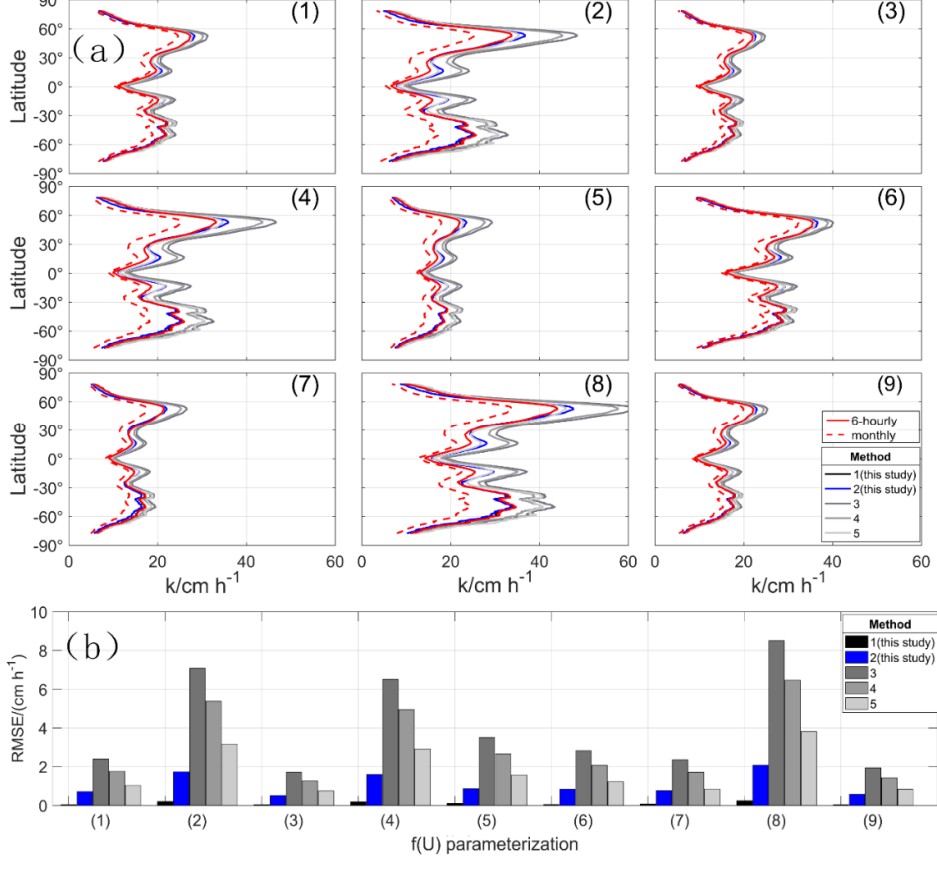

**Figure 7.** (**a**) Zonal profiles of corrected k for CO₂ using the five correction methodologies in comparison to annual k derived from 6-hourly (red solid curve) and monthly (red dashed curve) wind Scheme 1. (in black) is not visible because it overlaps with the 6-hourly k. Panels (**a1–a9**) show the latitudinal variations in nine k parameterizations listed in Table 1. (**b**) The RMSE of each method in corrected k from 6-hourly k.

Moreover, the relation between the averaging period $\Delta t$ (spanning from 6 hours to a month) versus $I_u^2$ (Figure 8), needed to infer biases in k computed using wind speeds at a coarse temporal resolution, was empirically derived and given as

$$I_u^2 = -0.18\Delta t^{-0.22} + 0.237. \tag{17}$$

Therefore, corrected f(U) in non-linear wind-only parameterizations for any gases at any temporal resolution from 6-hourly to monthly (or longer, not show here) can now be estimated by substituting $I_u^2$ in Equation (11) and Equation (14) for quadratic relations, and in Equation (15) and Equation (16) for cubic relations.

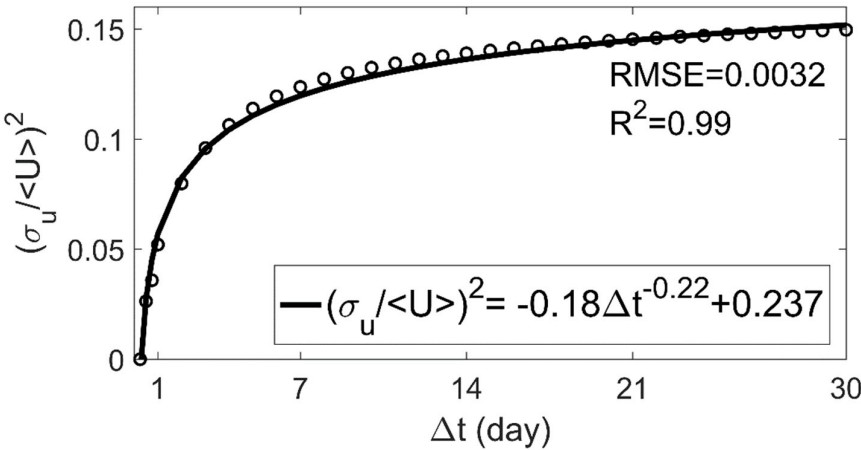

**Figure 8.** Coefficient of variation $I_u^2$ as a function of the averaging period $\Delta t$ (from 6-hourly to monthly). Circles indicate the results from measurements, and the solid line represents a modelled fit through the measurements. For $\Delta t > 18$ days, $I_u^2$ becomes independent of $\Delta t$. Global climate models operate on a $\Delta t = 30$ days.

### 3.4. Study Limitation

As shown by the Taylor expansion (Equation (9)), the bias due to time-averaging is a function of the variance in wind speed. Hence, a corollary question to explore is whether the 6-hourly wind speed used as a reference here introduces biases because of missing variances at sub-hourly time scales. There may be a fraction of energy in the scale of 6 hours to minutes or seconds commensurate with the time scales, over which the turbulent or wave action impact k but are presumably captured by the functional form of f(U).

The spectra of 6-hourly global wind is calculated to show the spectral energy distribution from mesoscale to decadal scale (Figure 9). As estimated, the variance in this range is $\sigma_d^2 = 0.1$ m$^2$ s$^{-2}$. A f$^{-3}$ scaling in the spectrum from multi-day to a 12-hour range appears to be supported here and implies an enstrophy cascade in quasi-geostrophic flow [58,59]. If the spectrum is extrapolated from 12 hours to turbulence scale (seconds) via a Kolmogorov's –5/3 power law, the "missing variance" in this range is $\sigma_m^2 \ll 0.001$ m$^2$ s$^{-2}$, which can be ignored. Extrapolations to finer scales via a f$^{-3}$ scaling would result in an even smaller missing variance. To be clear, this does not imply that the air turbulent time scales (on the order of 10 s and smaller) are minor. The energy contents of these time scales are quite large but are captured by the non-linearity in f(U) as noted earlier. To illustrate this point, a variance in turbulence scale $\sigma_t^2$ of 1 m$^2$ s$^{-2}$ can be estimated from a turbulence similarity relation $\sigma_t^2 = (2.3\,u_*)^2$ and $u_* = U\,C_D^{1/2}$, where $u_*$ is the air-side friction velocity and $C_D$ is a drag coefficient at the reference height of 10 m, derived elsewhere [60]. The energy content in turbulence is clearly an order of magnitude larger than that of the decadal to 0.5 hours timescale. However, the effects of these energetic eddies produce water-side eddies (or waves) that are captured by f(U). Therefore, it is safe to state from

this analysis that extrapolating the meso-scale variance to a sub-daily time scale introduces a negligible correction to k (Figure 9) provided the appropriate f(U) is used.

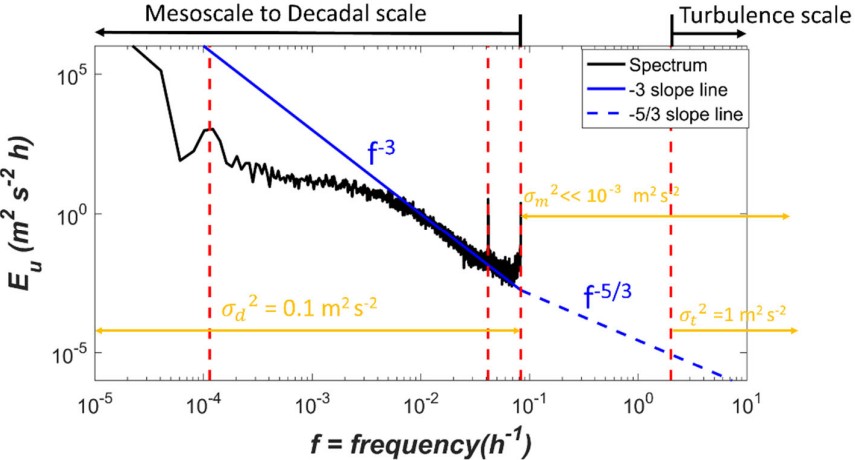

**Figure 9.** Energy spectrum of global average 6-hourly wind speed. The spectrum is extrapolated from 12 hours to a turbulence scale (seconds) via Kolmogorov's –5/3 power law ($f^{-5/3}$, blue dashed line). The resolved spectrum has an exponent of –3 from multi-day to 12 hours ($f^{-3}$, blue solid line) consistent with an enstrophy cascade in a quasi-geostrophic flow. The dashed vertical lines (right to left) indicate frequencies corresponding to the following timescales: sub-hour (=0.5 h), diurnal (=12 h), daily (=24 h), and annual (=8760 h), respectively. The $\sigma_d^2$, $\sigma_m^2$, and $\sigma_t^2$ refer to the variance at large (mesoscale to decadal), intermediate (12 h to turbulence), and small (turbulence) scales, respectively.

## 4. Conclusions

A new bias-correction model was developed based on a Taylor series expansion around the monthly mean wind state that accommodates any non-linearity in f(U) and explicitly considers wind variability effects on gas transfer velocity. This new model, expressed as a function of the squared coefficient of variation, $I_u^2$, can be used to adjust k and is derived relative to the 6-hourly time-averaged wind products. For the 6-hourly high resolution CCMP wind data, a simplified bias correction $k_b$ as a function of $I_u^2$ can be directly used to correct monthly k for any slightly soluble gas. With increasing wind variability over the last few decades associated with enhanced synoptic-scale high wind events [61,62], it is becoming increasingly necessary to quantify how the trend in wind variability biases k and influences global air–sea fluxes of $CO_2$ and other climate-relevant gases (e.g., $N_2O$, $CH_4$) in the next generation of climate models. In the absence of other wind statistics, a plausible approximation to correct monthly k is to set $I_u^2 = 0.15$. This correction can be readily accommodated in current climate models. More broadly, the moment expansion approach presented here can be adapted to correct biases associated with averaging non-linear functions so as to accommodate measurements at different temporal or spatial resolutions.

**Supplementary Materials:** The following are available online at www.mdpi.com/2072-4292/13/7/1328/s1, including Text S1, Tables S1 and S2, Figures S1–S3.

**Author Contributions:** Conceptualization, Y.G., G.G.K. and N.C.; Data curation, Y.G.; Formal analysis, Y.G., G.G.K. and N.C.; Supervision, G.G.K. and N.C.; Writing—original draft, Y.G.; Writing—review and editing, G.G.K. and N.C. All authors have read and agreed to the published version of the manuscript.

**Funding:** Y.G. is supported by a scholarship from the China Scholarship Council (CSC) under the grant CSC 201906710071. G.G.K. acknowledges support from the U.S. National Science Foundation (NSF-AGS-1644382, NSF-AGS-2028633, and NSF-IOS-1754893). N.C. is supported by the

"Laboratoire d'Excellence" LabexMER (ANR-10-LABX-19) and co-funded by a grant from the French government under the program "Investissements d'Avenir".

**Acknowledgments:** The cross-calibrated multiplatform (CCMP-V2) 6-hourly wind speeds are publicly available at www.remss.com/measurements/ccmp, accessed on 7 February 2021. The sea surface temperature data were obtained from https://cds.climate.copernicus.eu/cdsapp#!/dataset/reanalysis-era5-single-levels?tab=overview, accessed on 7 February 2021.

**Conflicts of Interest:** The authors declare no conflict of interest.

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
