# Peer review of "Mesoscale Temporal Wind Variability Biases Global Air–Sea Gas Transfer Velocity of CO2 and Other Slightly Soluble Gases"

_remotesensing, doi:10.3390/rs13071328_

Round 1

Reviewer 1 Report

This is a very interesting work in one of the more actual and important topics of geosciences. In fact, wind driven turbulence is the main driver of atmosphere-ocean transfer velocities, wind varies at micro-scales but Earth-system models, due to computational constrains, typically simulate at daily to monthly intervals and hundred to thousand Km wide cells. Therefore, corrections are mandatory, as well as testing and comparing different correction algorithms. I praise the authors for this work.

There is a fundamental clarification that is lacking and must be done as soon as possible in the text (preferably in the introduction): this and former correction methods are useless if wind variance is unknown. This is the case for Earth System Modellers and other end users. Therefore, in their cases, wind variance must also be supplied as input data. Or, at least, a sound coefficient of variation,

I am not enthusiast of using for finer resolution the 0.5º and 6hour data. Better resolutions are available for free in the Copernicus Marine platform. If not of global data, at least at regional scales. Is there some particular reason for having preferred this resolution, that the authors can share with us, and that may be interesting to better understand the work? The introduction and section 3.5 present good debates related with this. Maybe section 2.1 can present a brief argumentation for the choice of the finer resolutions?

The 5º spatial resolution grid was interpolated from finer resolution data, which is different from averaging over the data within each of the 5º squares, right? What was the interpolation method? If these are in fact different, the comparisons between 0.5º and 5º spatial resolutions to test the correction methods are not honest since, besides the method, the data in the background also changed, correct? Or did I miss something?

I have a problem with figure 2. I can easily see that averaging over 5º spatial dimension creates less bias than averaging over 1 month, since the wind variation should be smaller within 5º than within 1 month. What I don’t get is why coarser resolutions have opposite effects when averaging over space or over time. This contrast seems incoherent. Furthermore, it must be clarified what are the different k:

Is k_monthly a k estimated from 1 u value averaged over all u measured at 6h intervals?

Is k_6h a k averaged over all k estimated from all u measured at 6h intervals?

… and likewise for the k at different spatial resolution?

Maybe the topic could be better framed. Estimating transfer velocities from remote sensing the sea-surface does not always rely on wind estimates. As far as I know, no instrument on any satellite measures wind (correct?). What they measure is the radar scatter due to sea-surface roughness and relate it to wind. Some formulations for k make direct use of sea-surface roughness, which should be a better descriptor of the turbulence mediating k, albeit the uncertainty about which waves from the wave spectra should be used. Furthermore, the speed and direction of the wind relative to swell or peak waves is also important and may bring bias to remote sensing techniques. This could be better debated (still, briefly) than in lines 53 and 54 and references [1, 2 and 3]. At least briefly cite some of the related works by Godjin-Murphy, Woolf, Donelan, Chapron, Kudryavtsev, Soloviev, Landwher, Esters, the COARE team, etc. A reasonable review can be found by Vieira et al (2020) in MDPI\JMSE. References [1, 2 and 3] seem too short to me. This subject comes back again in lines 73-75. Although the introduction is generally well written, maybe this topic could be improved. Split among two parts of the introduction, one of them is one sentence (lines 53and 54) and the other is one small paragraph (lines 71-76), is not ideal (although I am not certain about how to do better). References in lines 73-75 could be attributed to each aspect rather than merged together in the end of the sentence. To me, it makes more sense merging the debate in lines 71-76 with lines 53-54. Furthermore, this way the debated in lines 55-70 proceeds directly to line 86; which makes much more sense than interrupting it with the small paragraph (lines 71-76). Figure 1 legend is unorthodox. The example should be in the body of the text and not on the legend. Figure 1 appears before its first mention in the text. Standard practice is the opposite. This mistake repeats with Table 1.

Wind is not the direct cause of transfer velocity, and its effect depends on the wind speed range and the sea-state. For low wind speeds, transfer velocity is basically mediated by micro-scale wave breaking, whereas for high winds, it is basically mediated by the turbulence and bubbles generated by the breaking of large waves. This is one of the reasons why there is no consensus on the correct exponent for wind-based formulations, even among open-ocean formulations. Obviously, this is a problem of the formulation itself, and not of the spatial and temporal resolution of the data and the averaging method, and thus is besides the scope of this work. Still, I wonder the bias brought about by averaging wind vs the bias brought about by using an inadequate formulation. Consequently, I also wonder whether the corrections proposed and tested are equally applicable and useful under all wind speeds, and under all sea-states. Are there useful comments that can be made in this work about these questions?

Not all open-ocean wind-based k formulations are either quadratic or cubic. Other exponents are also used in formulations that were neglected by the authors (see Vieira et al 2020 and the FuGas 2.5 for a thorough listing). This work would be more relevant if it presented a correction that could be tuned to any exponent. The equations for the corrections can be obtained empirically or solved analytically for other exponents besides 2, 3 or other integers. Taylor decompositions are applicable to functions with other exponents besides 2, 3 or other integers, converging in few terms. Applied to transfer velocity formulations, even complex physically-based ones, the Remainder becomes negligible after the first few terms are considered I recommend reading Vieira et al. (2013) for the application of the colocation polynomial and Newtons Finite Difference theorem for the numerical estimations of derivatives used in Taylor expansions applied to atmosphere-ocean gas exchanges. Therefore, I believe that the authors can develop and compare such more-generalist correction algorithms. I leave it as a suggestion for future work.

The presentation of the results is confusing. I had difficulty following the objectives of each section and how they differentiate each other. The reading of the results flows poorly. At the end, I still could not understand its logical sequence.

Section 3.2 \ Line 207-214 and figure (3): here, the bias was also averaged over 1990 to 2018. This must be clarified. So, this is the averaged bias due to averaging!!! The presentation starts to became confusing and I do not find this to be the best way for authors to pass their message. I suggest sending figure 3 to appendix and replacing by something less confusing than the averaged bias due to averaging; for instance, the traditional calibration-validation plot with estimates on the y axis and observations on the x axis.

Section 3.3: Reading the first sentence, it is supposed to be focused on comparing the current method with prior methods. However, its beginning gets confusing as it diverts into other aspects. One gets lost in this paragraph, and its tittle does not help clarification.

Figure legends are often insufficient. Keep in mind that an effective legend is the key to help a figure stand alone. The title should describe what the figure is about. The methods include all (and no more than) the necessary details to understand the figure without referring back to the body text.  This does not happen in this submission, where the figure legends are sometimes poor.

Miscellaneous editing issues:

The averaging operation being first represented by overbars, and later replaced by brackets, does not work.  It only brings confusion and halts the flow of the reading , demanding for an extra dose of concentration, that should be unnecessary.

Lines 117-118: first “(CCMP)-v2” and then “CCMP-v2”.

Line 145: needs a reference.

Line 171 Equation 12: brackets are missing, representing the averaging operation.

Line 181 – Section 3.1 heading: this heading is incorrect since this section also addresses the bias induced by spatial averaging.

The article cannot have a heading on line 181 saying:

“Bias in k induced by temporal averaging of wind speeds”

And then, in line 182, the first sentences in the paragraph says:

“averaged k for CO2 at various temporal and spatial resolutions are assessed”

“oral and spatial resolutions are assessed for the parameterizations of f(U) listed in Table 1. The k computed from maximum spatial (0.5°×0.5°) and temporal resolution (6-hourly) products are used”

Figures 2 and 5- y axis label: Clarify what is Delta k %. My guess is that it is % of k i.e., (Delta k)/k*100.  But this must be explicit.

Figure 3 legend – be explicit to what the bias refers: k_monthly-k_6h? R2? R3? Term1-term2?

Line 204-207: Maybe this could be stated more clearly. I suggest splitting the sentence in two.

Line 221: very confusing sentence. Split in smaller sentences is highly advised. At least, insert a missing comma in “… , and averaged kb (i.e., with averaged …).

Reviewer 2 Report

Good work, describing algorithms for improvement errors in computation of the air-sea  gas transfer velocity induced by non-linearity of the wind impact.    Authors compared 9 different formulas for gas transfer velocity (k). Influence of the spatial and temporal averaging are discussed. Results of the work can be used for climatic and ecosystem modelling.

Some remarks:

  • For my opinion - better add graphical presentation of the used parameterizations (see attach)
  • Some additional explanations need for SST non-linearity description.
  • It’s not correct ( from physical point of view ) to convert eq 2 into F(U) = k*(Sc/660)5
  • Pls explain - Why wind from CCMP-V2, but SST from ECWMF ERA5? ( for publishing in Remote Sensing?) For wind used reasonable use satellite data (GHRSST?), or for SST ECMWF wind from same source?
  • What is it – standard formulation for SST (line 130)? Reference [35] refers [25] for this formula.

Author Response

This manuscript is a resubmission of an earlier submission. The following is a list of the peer review reports and author responses from that submission.

Round 1

Reviewer 1 Report

Please see a separate pdf file with specific comments, questions, and suggestions.

Reviewer 2 Report

Comments on remotesensing-1072725 entitled "Mesoscale Temporal Wind Variability Biases Global Air-Sea Gas Transfer Velocity of CO2 and Other Slightly Soluble Gases" by Yuanyuan Gu, Gabriel G. Katul, Nicolas Cassar.

This paper presents the new correction method considered with wind variability for the estimation of global air-sea CO2 gas transfer velocity. The effect of the wind variability is considered to be important, since the wind speed fluctuates more than other parameters. However, this study is not described adequately for all sections. This paper would need major revisions before being suitable for publication.

Major comments

(1) Although this paper is very simple in scope, I found it difficult to follow because the structure of this paper is inadequate, of which there are many. (The table shown in section 2 is used in section 1, there is no description of the method in section 2, but the method is described in the results section. Section 4 is discussion and there is no conclusions. etc.)

(2) In the Introduction section, the authors should clearly mention their objective and the drawbacks of the previous methods.

(3) The used CCMP-v2 wind data in this study is not mentioned. The authors also need to describe the details of the CCMP-v2. And there is no description about the monthly data of wind speed. In addition, most of the global wind speed data is provided by 6 hourly.  Consequently, I do not understand why the results of using 6 hourly and monthly datasets were compared.

(4)In Table 1, No. 9 lists three models, but only one model is used. The authors should clearly mention the reason.

(5)The authors sets Iu2=0.15. The authors should mention the accuracy of CCMP wind speed and consider the accuracy.

Reviewer 3 Report

Authors present the research aimed at correction of biases introduced into data processing that involves averaging and non linearity in the processing steps.

With the increasing amounts of data available to researchers and a need invoke data averaging at different steps of processing the question of correction of bias introduced by averaging is important.

I find the research appropriate and valuable. I would like to mention that the paper may be interesting not only to researhers dealing with gases transfer across sea-atmosphere border, but also to all researchers dealing with any type of averaged data and non-linearity.

I only would like to mention one mistake (typo) on line 112:

Perhaps it should be "can BE used for other"

Reviewer 4 Report

The authors are trying to apply parametric formulas for gas transfer velocity based on standard 6-hour wind data for longer temporal scales. The authors see the main reason for the emerging bias in the non-linearity of the parametric formulas. The polynomial powers in the cited formulas are not higher than cubic and the corresponding corrections can be expressed in terms of dispersions and skewness of the distribution of wind velocities. Calculating these corrections directly (grid-by-grid option of Table 2, line 1) or assuming the fixed variance coefficient I2u=0.15 the authors present deviations of the corrected gas transfer velocities from the 6-hour counterparts.

The referee does not see a novelty and clear physical idea in developing corrections for monthly and annually-averaged data starting with 6-hour data that accounts for short-scale physical processes.
The paper is poorly written. English and terminology require substantial corrections.
The referee cannot recommend the paper for Remote Sensing.

Some particular remarks are listed below:

1. The abstract is full of mathematical notations but some of them are not properly introduced. E.g. line 15, what is f? Is it k?
2. Eq. 1 - What is F? Why k and K0 are in brackets?
3. L.48 "Because these mechanisms require some averaging time". Physical mechanisms never require "averaging time". Please, re-phrase;
4. Eq.2 looks illiterate for a physicist because it relates two quantities with different dimensions. Thus, the coefficients of the parametric formula are dimensional that implies their dependence on undefined physical scales;
5. L.99. Why the authors need SST (sea surface temperature, unfortunately, the abbreviation is not defined in the paper in a special section of the MDPI template)? There is no dependence on temperature in Table 1;
6. Eq.8. What does it mean <f(U)≥? Please, use conventional hyphenation in the mathematical expression;
7. L.236-237. The effect of SST on k has not been analyzed before this line (see also point 5);
8. L.257-258. The Taylor series cannot show that something is a function of variance. It is your hypothesis, very questionable, by the way;
9. Line 271. "These... these" Please, correct;
10. Line 274 "air side friction velocity" a bit strange word combination.